# Attitudes, Behaviors, and Perceptions of Students Vaping in Three Mexican Universities

**DOI:** 10.3390/ijerph21040464

**Published:** 2024-04-10

**Authors:** Genny Carrillo, Nina I. Mendez-Dominguez, Maria Elena Acosta Enriquez, Javier Morán-Martínez

**Affiliations:** 1Department of Environmental and Occupational Health, School of Public Health, Texas A&M University, 212 Adriance Lab Road, College Station, TX 77843, USA; 2Hospital Regional de Alta Especialidad de la Peninsula de Yucatan IMSS-BIENESTAR, Calle 7 #433 X20 y 22 Fracc, Altabrisa, Mérida 97130, Yucatán, Mexico; ninamendezdominguez@gmail.com; 3Sciences of Health Faculty, Montemorelos University, Libertad 1300 Pte., Montemorelos 67500, Nuevo Leon, Mexico; elenamaria@um.edu.mx; 4Facultad de Medicina, Departamento de Biología Celular y Ultraestructura, Universidad Autónoma de Coahuila, Avenida Morelos 900 Oriente, Primero de Cobian, Centro, Torreón 27000, Coahuila, Mexico; javiermoranmartinez@uadec.edu.mx

**Keywords:** vaping, Mexico, cross-sectional studies, universities, students

## Abstract

The increase in the popularity and use of electronic cigarettes (e-cigarettes) has consistently risen worldwide and has become associated with adverse health outcomes. This study has identified the attitudes and perceptions of undergraduate students who vape in three universities in Mexico. A cross-sectional study involving 495 participants was conducted using a survey from October to December 2023. Three universities in different states in Mexico collaborated with colleges in Yucatan, Durango, and Nuevo Leon, Mexico. Descriptive statistics include frequencies and percentages, and four logistic regression models were employed. In the sample, 31% and 17.54% of participants reported smoking and vaping, respectively. Students who reported vaping in the last month reported that their first experience with vaping was at an average of 17.3 years of age, and of those, 71.26% (n = 62) reported having vaped for over 100 days, while the remaining 38% reported vaping for between 2 and 100 days. Students from Veracruz and Yucatan began vaping at a younger age than in the central and northern regions. There is a need to educate students about the dangers of the chemicals in the liquids they use, the secondary exposure vapers, and the health dangers they pose.

## 1. Introduction

The increased worldwide popularity of e-cigarettes surpasses the speed that scientific research and learning must progress at due to the long-term health effects of them [1]. E-cigarettes are believed to be an alternative to traditional cigarettes according to evidence produced in the findings of two randomized controlled trials indicating that e-cigarettes with nicotine can help adults who smoke stop smoking in the long term compared with placebo (non-nicotine) e-cigarettes [2,3]. However, the research was inconclusive, and the Food and Drug Administration (FDA) has not found any e-cigarette to be safe and effective in helping smokers quit [4]. E-cigarette use has been on the rise, especially among young adults, adolescents, and children in the U.S. In 2023, 2.13 million U.S. middle and high school students stated using e-cigarettes in the past 30 days, comprising 4.6% of middle school and 10.0% of high school students [5]. The electronic nicotine administration system (ENDS) includes vapes, vaporizers, vape pens, e-cigarettes, e-cigars, and e-pipes. Most e-cigarette liquids contain nicotine, which is addictive and highly toxic to humans and developing fetuses in small doses. Its use has been associated with many adverse health outcomes like throat and mouth irritation, a dry cough, nausea, decreased cardiovascular function, asthma, and lung injury [6,7]. The fatal dose of nicotine has been estimated at 30–60 mg for adults and around 10 mg for children. When nicotine is inhaled, it reaches the bloodstream and is transported to the brain, binding with nicotine acetylcholine receptors, and in e-cigarettes, nicotine is in a freebase that vaporizes quickly [8]. Nicotine salts used in the e-cigarette are formed by adding benzoic acid, usually helping vaporize it at a lower temperature, causing better absorption by the body because they are less volatile and produce an instant rush of nicotine [9]. The nicotine salts used in e-cigarettes vaporize at a lower temperature, which results in more efficient absorption by the body because they are less volatile [10].

A Centers for Disease Control and Prevention (CDC) study found that 99% of the e-cigarettes sold in the United States contained nicotine [11]. As the long-term effects of e-cigarette use remain unknown, the widespread use of these devices poses a severe ongoing public health concern. Studies report that many individuals initiate tobacco use as young adults (38% of U.S. adults aged 18–24 years old are enrolled in colleges) [12]. This makes colleges and universities a critically important environment in which to provide smoking cessation support, prevent e-cigarette use initiation, and prevent these young adults from developing symptoms and medical conditions associated with the use of e-cigarettes [13]. 

Although there is currently a lack of knowledge on the long-term effects of exposure to e-cigarettes, research has found that the 7700 flavors in 400 brands of e-cigarettes available contain several chemicals such as a flavoring compound used in food products and diacetyl, which, when heated, will transform into harmful chemicals causing respiratory irritations [10]. Many flavoring compounds used in vaping liquids are considered safe or are “generally recognized as safe” (GRAS) for oral consumption by the FDA. However, it has been found that prolonged inhalation of some GRAS flavorings, such as diacetyl, 2,3-pentanedione, and acetoin, can cause permanent lung injury and disease [14].

As of 1 July 2023, 2614 colleges and universities in the USA had adopted smoke and vape-free campus policies [15]. Still, despite this progress and support from campus communities and the tobacco-free policy that has resulted in decreased smoking rates, compliance rates continue to pose a challenge [16]. Unfortunately, the marketing strategies used by e-cigarette manufacturers and distributors that have supported the notion that e-cigarettes are less harmful, aid effective smoking cessation efforts, and are a safe alternative to traditional cigarettes have led to an overall low level of risk perception associated with e-cigarettes and a tendency to follow tobacco-free campus policies [17].

### Mexico Policies

Mexico’s situation is different since in 2012, the Federal Commission for Protection against Health Risks (COFEPRIS) announced a prohibition on the sale of electronic nicotine delivery systems (ENDSs) throughout the country, among which devices such as e-cigarettes and related accessories stand out [18]. 

In 2022, article 16 of the Mexican General Law for Tobacco Control, section VI, prohibited “the manufacture, import, distribution, promotion, and marketing of any object that is not a tobacco product and contains any of the elements of the brand or any design or sign that identifies it with tobacco products, as is the case with electronic cigarettes” [18].

Despite this legal prohibition, the distribution of e-cigarettes in the country has expanded rapidly, with such devices being illegally sold to more than one million adults and nearly 300 thousand adolescent and young consumers [19,20]. However, in May 2022, Mexico President Andres Manuel Lopez Obrador signed a decree prohibiting the circulation and marketing, regardless of device origin, of ENDSs, similar systems without nicotine, alternative nicotine consumption systems, electronic cigarettes, and vaporizing devices and their associated solutions and mixtures within the Republic of Mexico [21].

COFEPRIS reported in 2023 that at least five million people between 12 and 65 years of age had used a vape pen, and the first use of electronic cigarettes that contain nicotine and other substances, such as cannabinoids, occurred at an average of 12 years of age [22]. This study provides critically needed evidence that identifies the attitudes and perceptions of undergraduate students who vape and those who do not in three universities in Mexico.

## 2. Materials and Methods

### 2.1. Study Design, Participants, and Location

A cross-sectional study was conducted using a survey from October to December 2023. Three universities in different states in Mexico collaborated to recruit undergraduate students from different colleges in each university in the states of Yucatan, Durango, and Nuevo Leon.

### 2.2. Recruitment and Study Population

The student sample was composed of higher education students living in Mexico as students enrolled in academic institutions, both private and public. Students from each university voluntarily signed up for this study and completed a survey online. The study population included groups by university location and vaping status, also used as a variable for comparison. The inclusion criteria for this study consisted of college students aged 18 and above who were enrolled in classes offered by the participating Mexican universities and completed the whole questionnaire. The exclusion criteria were being unable to respond to the questionnaire; individuals who were not enrolled at participating universities and those who did not complete the questionnaire were eliminated. 

The Institutional Review Boards (IRBs) of Texas A&M University (IRB 2023-1179), the Universities of Montemorelos (Reference No. 2023/145), and the University Autonomous of Durango (Reference No. 2010/23), Mexico, reviewed and approved the research protocols used in this study. 

### 2.3. Data Collection

The survey was delivered using a Google Form, providing participants with a convenient and user-friendly platform for submitting their responses. The questions included were designed to collect demographic and self-identification information from respondents, information about their educational status and affiliations with organizations on their university campus, and questions related to their smoking habits and e-cigarette use that offered insights into participants’ past and recent tobacco and nicotine product usage. 

The questionnaire was validated for comprehension and readability using a pilot study with 36 university students from a single university who responded to a ten-item survey instrument. We obtained a Cronbach’s alpha of 0.76 for validity and a scale reliability coefficient of 0.86. Based on the feedback from the pilot test pilot, we added the source for obtaining vaping devices and accessories and re-sellers based on respondents’ suggestions.

The survey instrument included questions about participant vaping status: never vaped, former vapers, and current vapers (those who have vaped at least weekly in the past month). For newer vapers, questions were designed to gain a deeper understanding of participant curiosity and intentions and whether there was peer influence related to e-cigarette use among individuals who had not previously used e-cigarettes. Individuals with past e-cigarette usage (defined as those who had not vaped in the last month) were asked questions intended to gather information about their age of initiation, reasons for using e-cigarettes, and the regularity of e-cigarette use. Lastly, for current vapers, the questions focused on their e-cigarette usage and experiences, including reasons for current e-cigarette use, rate of use in the past 30 days, sources of e-cigarette product purchases, types and brands of e-cigarettes used, flavored e-cigarette use, awareness of nicotine content and nicotine salts, intentions to quit, past attempts at quitting, their usage patterns, cravings, the difficulties they encountered in restricting e-cigarette use, and their reactions when they could not use e-cigarettes such as feelings of irritability, restlessness, and anxiety. Those questions were intended to identify essential perceptions of the behaviors and experiences of individuals who currently use e-cigarettes. Additional questions were asked to assess participant interest in receiving educational materials and resources at their respective universities about the health consequences of vaping and if they knew that using ENDSs was prohibited in Mexico.

### 2.4. Statistical Analysis

Descriptive statistics describing the respondents’ place of residence, sex, occupation, parental and sibling education level, and type of academic institution are presented as total percentages or means depending on their nominal or numeric nature. Three groups were considered, according to the region where their universities are located and the sample size for each cluster, with a contrast of independent proportion sample size, with a confidence of 95% and a maximum error of 0.05. n = (Zα/2 + Zβ)^2^ × (p1(1 − p1) + p2(1 − p2))/(p1 − p2)^2^ = 165 per cluster (495 individuals).

The prevalence of ever having smoked or vaped and having smoked or vaped in the last month is presented with grouped sociodemographic variables. Four logistic regression models were employed to obtain measures of association between (1) vaping in the last 30 days, (2) ever having tried to quit smoking or vaping, (3) symptomatology related to smoking or vaping, and (4) planning to quit smoking. All statistical analyses were performed using Stata version 14.0.

## 3. Results

Among the 495 respondents to the survey, 289 were women, 172 were men, 20 were non-binary, 5 preferred not to answer the question, and 9 left the item blank. Regarding ethnicity, 326 participants identified themselves as Latino or Hispanic, 36 did not self-identify with a specific ethnicity or origin, 38 considered themselves mestizos, 12 of non-Hispanic European origin, 5 of Mayan or Mayan descent, and 2 of African descent, with 77 respondents not specifying an ethnicity. All participants were students, but 82 reported being employed full-time while studying, and 124 reported being employed part-time. A total of 390 respondents reported living with their family and 5 with relatives, 47 reported living with roommates, 44 reported living alone, and 10 reported living with a couple or significant other.

Of the 495 participants responding to the study, 250 had smoking antecedents, 154 reported smoking, and 87 reported vaping in the last month. Of the 154 respondents who reported smoking, 119 considered it probable to highly probable that they would try vaping soon. Overall, 31% and 17.5% of respondents reported that they smoked or vaped, respectively. Students who reported vaping in the last month reported that they first vaped at a mean age of 17.3 years; 27 students reported that they started vaping because a friend vaped, 5 because a relative vaped, 10 because they were trying to quit smoking cigarettes, 1 because the student thought they are less harmful than cigarettes, 1 because they saw people using them on social networks, movies or tv, 8 because the student knew they were available in several flavors, 28 due to curiosity, 8 due to anxiety, depression, or sadness, 4 because they are easier to use without being noticed, 2 because they wanted the experience of the effect of nicotine, and 1 because they could learn tricks to entertain their friends.

A total of 71.26% (n = 62) of respondents reported over 100 days of vape use, while the remaining 38% reported vaping for between 2 and 100 days. A total of 14 students responded that they bought vaping supplies and accessories from the internet, 55 from a reseller in their network, and others mentioned buying them in retail stores or from unspecified sources. Sixty-seven students reported using disposable e-cigarettes, twelve used rechargeable vaping devices, three reported using liquid containers, and five did not specify the ENDS device they used. Of the 87 respondents who reported that they used vaping devices, 85 reported using flavored vaping solutions, of whom 50 preferred mint flavor. A total of 83 respondents reported that they were aware of the presence of nicotine in their vaping solutions, 63% knew about nicotine salts included in vapers, and 76 consumed vaping solutions with a concentration of between 2 and 5% nicotine. In contrast, 11 reported using vaping solutions with a nicotine concentration above 5%.

Forty-five participants reported that they were currently thinking about quitting vaping, of whom six reported that they intended to quit in the next 12 months. A total of 4 reported that they were planning to do so in the next 6 months and 18 in the next 30 days, with 14 reporting that they had no plans to quit vaping. Sixty respondents reported that they had tried to quit vaping in the last 12 months between three and ten times, while eighteen respondents reported that they vaped over 30 times a day, and twenty-two participants reported vaping zero to four times a day for at least 10 min.

Forty-four respondents reported vaping less than an hour after waking up, and thirteen reported that they woke up at night to vape, ten of which reported waking up to vape three or more nights a week. Twenty-seven participants reported that it was difficult for them to vape in places because it is illegal. Twenty-two participants responded that they felt irritable or anxious when they could not vape freely while trying to quit smoking. Sixty participants reported having symptoms related to vaping, with upper respiratory tract irritation being the most commonly reported, but also dizziness, headaches, and chest congestion. See Figure 1.

When analyzing by groups, age was a factor in those who smoked and those who had not, though respondent sex was not found to be a factor differentiating cigarette or vaping users. We found that students in the southern Mexican states (Veracruz: 15.25; Yucatan: 17.07 on average) reported a younger age of first smoking than those in the central and northern regions of Mexico (17.8 on average).

As shown in Table 1, we found that age, sex, ethnicity, and type of higher educational institution were not factors that affected propensity to vape. However, individuals self-describing as non-binary were found to be more likely to vape, and those who worked or lived with family were not more likely to have vaped in the last month. Being a university student from a family with no higher education was significantly associated with reduced odds of vaping in the last month, as was attending a private rather than a public institution of higher learning. Smoking cigarettes in the last month increased the likelihood of a participant having vaped in the last month, with a *p* < 0.001.

In our Poisson regression model, we tested factors linked to attempts to quit smoking (Table 2) and found that age and symptom distribution (upper respiratory symptoms, irritability, nervousness, chest oppression, or difficulty breathing) did not vary regarding the number of attempts, not even in the presence of one or more symptoms; those who had made more attempts were those who woke up at night to vape and those who found it challenging to not vape where it is prohibited.

Regarding symptomatology (Table 3), the presence of symptoms among those who vaped in the last 30 days was associated with sex, with women being the most affected, but also with those who were first-generation university or college students. We also found a dramatically increased risk among cigarette smokers.

Finally, based on the correlates related to planning to quit vaping in the next month and the next twelve months (Table 4), we found no differences related to age or sex. Still, we did find that nicotine users were more prone to plan to quit vaping, along with those who had started vaping at an older age and those who had tried to quit vaping before.

## 4. Discussion

The results from the survey of students from universities in Mexico showed that the number of women (289) vaping were higher than that of men (172), which is different from a study reporting that the prevalence of e-cigarette use in male adolescents was consistently higher than among females, while another reported that females vaped more frequently than males [23,24,25]. Most of the participants in this study lived with their families, while only 44 lived alone. Commonly, students who live with their families and extended families rely heavily on them for support [26]. Our study’s estimated mean age of e-cigarette initiation was 17.3 years, consistent with a study that reported 17.50 years of age (95% CI = 17.47, 17.52) [27,28,29,30].

In our study, 76 students reported that they use an e-cigarette with a nicotine concentration between 2 and 5%, while 11 reported using a vape liquid with a nicotine concentration over 5%. However, vaping behavior strongly affects nicotine uptake among individuals and depends on factors such as puff duration, intensity, and frequency, as well as if they are using nicotine salt e-liquids that are designed to deliver nicotine more efficiently and smoothly compared to traditional freebase nicotine e-liquids. Many studies have found that the level of nicotine exposure from each puff is highly variable due to the variability in aerosolization [31,32]. For some vapers, especially if they were former heavy smokers with a high level of nicotine dependency, a 20 mg/mL concentration might be sufficient to satisfy their nicotine cravings. Taylor et al. studied the accuracy of the levels of nicotine in e-cigarette liquids and found that nicotine strength was commonly mislabeled. Upon analysis, the concentration was higher than listed on the label, and on one occasion, a nicotine concentration of more than 20 mg/mL was found in a product labeled as nicotine-free [8].

Individuals who are beginning to vape and use an e-liquid with a strong nicotine concentration can potentially experience adverse effects like dizziness, headaches, or nausea, as was reported by 60 participants in our study, and upper respiratory tract irritations, as have been reported in other studies [33]. A case was published involving a healthy 31-year-old Mexican male who reported daily e-cigarette use that contained both nicotine and THC [34]. A patient from Mexico with vaping-associated lung damage and seizures met the CDC definition of e-cigarette-causing lung injury (EVALI) [7].

A study by Morean et al. reported that individuals do not understand the nicotine information on e-liquid labels because the concentrations are presented using mg/mL and percentage of nicotine. E-liquid consumers are unable to translate these metrics into an estimation of nicotine strength accurately. The researchers explained that to convert the percentage of nicotine to mg/mL, one must multiply by 10 (e.g., 5% nicotine = 50 mg/mL) [35].

Forty-four of our students reported that they vaped less than one hour after waking up, and thirteen reported waking up at night to vape, which is what Branstetter et al. reported in their research where his participants were waking at night to smoke as a possible result of nicotine dependence and psychological distress [36]. Our participants reported that they began vaping because their friends or a relative did it, were vaping as part of an effort to quit smoking, because vaping was perceived as less harmful than cigarettes, saw the marketing of these products in social networks, because of the flavors, out of curiosity, because of tricks they could perform to entertain their friends, because of depression or sadness, or due to being desirous of experiencing the effects of nicotine. Another study reported that college students use e-cigarettes primarily for enjoyment [37].

Being a first-generation university student and attending a private educational institution were significantly associated with reduced odds of vaping in the last month. Those results differ from those of the study by Saima et al., which reported that in Karachi, about 52% of the participants who vaped attended a private university [25].

The limitations inherent to this study include sources of error and biases that influence the wording and format of the instrument’s questions, and the fact that survey data cannot provide robust evidence of cause and effect. However, the risk of bias was minimized by including (a) random, (b) sufficient, and (c) representative samples.

## 5. Conclusions

This study provides information about the attitudes, behaviors, nicotine use, and preference for flavored e-juices of student vapers, the reasons why they began vaping, the presence of symptoms related to vaping reported by the respondents, and the difference in vaping between private and public university students in Mexico. We found that study respondents reported being frustrated that they were not able to vape at their universities and that there were no policies in place at their campuses to provide education or enforce campus vaping bans. We also found that the current ban on the sale of vaping products put in place by the Mexican government has increased the purchasing of vaping products both online and on the black market, which makes it more difficult for users to purchase quality, trusted products with an accurate amount of nicotine reported on product labeling. There is a need to educate students about the dangers of the chemicals in e-cigarette liquids, as well as the health consequences of secondary exposure to vapors for those in proximity to those who vape.

## Figures and Tables

**Figure 1 ijerph-21-00464-f001:**
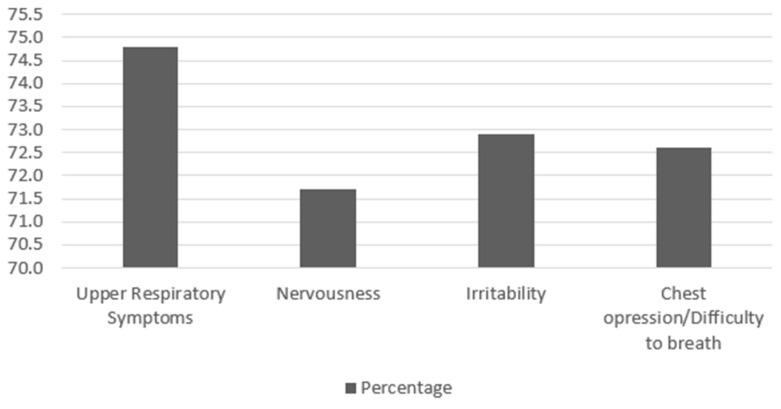
The symptoms reported by students who reported smoking in the last 30 days (n: 87).

**Table 1 ijerph-21-00464-t001:** Association of vaping in the last month with the participants’ demographics.

Vaping in the Last Month	Odds Ratio	Std. Err.	z	*p*	[95% Conf. Interval]	
Private institution	0.62	0.12	−2.39	**0.017**	0.42	0.92
Age	1.019	0.030	0.64	0.520	0.96	1.08
Career or university	1.27	0.89	0.34	0.736	0.32	5.02
First person obtaining a degree in the family	0.59	0.152	−2.05	**0.040**	0.35	0.97
Gender ratio male/female	0.97	0.20	−0.15	0.884	0.64	1.45
Non-binary	2.58	1.05	2.34	**0.019**	1.16	5.72
Ethnicity self-identified	1.22	0.30	0.82	0.414	0.75	1.98
Job	1.127	0.23	0.59	0.554	0.76	1.68
Cohabitation with family	1.26	0.314	0.95	0.342	0.78	2.06
Smoking cigarettes in the last month	15.00	4.16	9.76	0.000	8.71	25.8

**Table 2 ijerph-21-00464-t002:** Poisson regression model for attempts to quit smoking.

Attempts to Quit Smoking	Incidence Rate Ratio	Std. Err.	z	*p* > z	[95% Conf. Interval]	
Age	0.96	0.03	−1.43	0.154	0.90	1.02
Gender	1.16	0.15	1.14	0.254	0.90	1.49
Wakes at night to vape	1.95	0.34	3.82	**0.000**	1.38	2.75
Finds it challenging not to smoke where it is prohibited	1.28	0.16	2.05	**0.041**	1.01	1.63
Symptoms	1.10	0.15	0.7	0.481	0.84	1.44

**Table 3 ijerph-21-00464-t003:** Association between symptoms and smoking cigarettes, gender, and family.

Symptoms	Odds Ratio	Std. Err.	z	*p* > z	[95% Conf. Interval]	
Cigarettes in the last month	6.05	4.66	2.34	0.019	1.34	27.30
Women	4.46	3.14	2.13	0.033	1.12	17.71
First in the family to study	0.15	0.12	−2.25	0.024	0.03	0.78

**Table 4 ijerph-21-00464-t004:** Association between planning to quit vaping and age, gender, nicotine, and symptoms.

Plans to Quit Vaping in the Next Month to 12 Months	Odds Ratio	Std. Err.	z	*p*	[95% Conf. Interval]	
Age	0.91	0.20	−0.43	0.67	0.59	1.40
Gender	0.45	0.33	−1.08	0.282	0.10	1.93
Age when first vaped	1.39	0.20	2.35	**0.019**	1.05	1.84
Nicotine use	3.04	1.67	2.00	**0.045**	1.02	9.03
Symptoms	0.91	0.68	−0.12	0.904	0.21	3.95
Previous attempts to quit	1.34	0.148	2.62	**0.009**	1.07	1.66

## Data Availability

The data used for this study are available upon request.

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
