# Peer review of "Attitudes, Behaviors, and Perceptions of Students Vaping in Three Mexican Universities"

_ijerph, 2024, doi:10.3390/ijerph21040464_

Round 1

Reviewer 1 Report

Comments and Suggestions for Authors

I recommend that authors enhance the quality of their work and re-submit.

Below suggestions may help.

Introduction:

Line 32, ‘’still new’’, why this is added here?

Line 39, ‘’Food and Drug Administration (FDA)’’ and line 68 ‘’Food and Drug Administration (FDA)’’. Use the abbreviation is line 68

Line 55, ‘’CDC’’. Use the full name before first abbreviation.

Line 99, ‘’in said systems’’, change to: above mentioned systems

What is the difference between Electronic Nicotine Administration Systems, Similar Systems without Nicotine, Alternative Nicotine Consumption Systems and ‘’Nicotine replacement therapies- NRTs’’.

Line 88, COFEPRIS, mention the full name before the abbreviation. The full name also is mentioned in line 100

Methods:

Line 139: current vapers (those who had vaped in the past 30 days). Did you consider participants who if even vaped once, or took a puff, as current vaper? Current vapers should be defined clearly

The broad inclusion and exclusion criteria made the included groups vague.

Have the authors done sample calculations?

There is no clear definition to each group included.

I assume that the no criteria for exclusion criteria

Results:

It is not clear which participant were former smokers and currently vapers?

I assume in a such study, groups should be: pure vapers, vapers but ex-smokers, cigarette smokers-if included, former vapers .. etc.

The way how the results were reported is not clear. The whole results section needs to be re-written in a clearer way.

Comments on the Quality of English Language

Must be enhanced

Reviewer 2 Report

Comments and Suggestions for Authors

The study topic is important, but this study requires extensive revisions.

1/ There is a high percent of similarity of the text with other studies - see iThenticate report

2/ More precise data on the study group should be added

3/ Map can be removed

4/ The study sample size is low. Please justify.

5/ Please revise the title to clearly define the study population

6/ Please provide more advanced data in the results section

7/ Please assess how the study population and the sample size impact the results and the risk of bias

Comments on the Quality of English Language

Please revise the text to reduce the % of similarly of the text with other papers

Reviewer 3 Report

Comments and Suggestions for Authors

The aim is of the research is very important for the country Mexico but don’t have the crucial impact for international knowledge about vaping. However still is worth of publishing as comparision for other countries.
The study was designed in a typical, the most popular way. In that case the study design didn’t bring something new to the researches. However the design of study was prepared in a proper way. The study fills a research gap in the area of knowledge about vaping in Latin America.

In a chapter 2. Materials and Methods, 2.1. Study design, participants, and location, line 109 - what does "mixed" methodology mean? The description in data collection shows that there was only a questionnaire in Google Forms. What do they mean “mixed”? What other tools are used with google forms?

Conclusions are relevant to the aim of research and answered the gap.
References are relevant.

Round 2

Reviewer 1 Report

Comments and Suggestions for Authors

 Results:

Please recheck the sample size reporting. It was 496 in the first draft, in the revised version 495, when I calculated the number, it was 486 in the first section (289 were women, 172 were men, 20 were non-binary, and five preferred not to answer the question.)

Reviewer 2 Report

Comments and Suggestions for Authors

The manuscript was significantly improved and can be considered for publication in IJERPH.
